# Enhanced Low-Neutron-Flux Sensitivity Effect in Boron-Doped Silicon

**DOI:** 10.3390/nano10050886

**Published:** 2020-05-05

**Authors:** Guixia Yang, Kunlin Wu, Jianyong Liu, Dehui Zou, Junjie Li, Yi Lu, Xueyang Lv, Jiayun Xu, Liang Qiao, Xuqiang Liu

**Affiliations:** 1Key Laboratory of Radiation Physics and Technology of Ministry of Education, Institute of Nuclear Science and Technology, Sichuan University, Chengdu 610064, China; biansechong@163.com (G.Y.); xjy@scu.edu.cn (J.X.); 2Institute of Nuclear Physics and Chemistry, China Academy of Engineering Physics, P.O. Box 919-220, Mianyang 621900, China; klwu334967027@foxmail.com (K.W.); m13568266352@163.com (J.L.); zdhahuier@sina.com (D.Z.); lastljj@hotmail.com (J.L.); luyi3060@163.com (Y.L.); lvxueyang04@163.com (X.L.); 3School of Physical Electronics, University of Electronic Science and Technology of China, Chengdu 610054, China

**Keywords:** enhanced low-neutron-flux sensitivity effect, noise power spectral density, carrier removal rate, remaining vacancy-related defect, collector-emitter leakage current

## Abstract

Space particle irradiation produces ionization damage and displacement damage in semiconductor devices. The enhanced low dose rate sensitivity (ELDRS) effect caused by ionization damage has attracted wide attention. However, the enhanced low-particle-flux sensitivity effect and its induction mechanism by displacement damage are controversial. In this paper, the enhanced low-neutron-flux sensitivity (ELNFS) effect in Boron-doped silicon and the relationship between the ELNFS effect and doping concentration are further explored. Boron-doped silicon is sensitive to neutron flux and ELNFS effect could be greatly reduced by increasing the doping concentration in the flux range of 5 × 10^9^–5 × 10^10^ n cm^−2^ s^−1^. The simulation based on the theory of diffusion-limited reactions indicated that the ELNFS in boron-doped silicon might be caused by the difference in the concentration of remaining vacancy-related defects (*V*_r_) under different neutron fluxes. The ELNFS effect in silicon becomes obvious when the (*V*_r_) is close to the boron doping concentration and decreased with the increase in boron doping concentration due to the remaining vacancy-related defects being covered. These conclusions are confirmed by the p^+^-n-p Si-based bipolar transistors since the ELNFS effect in the low doping silicon increased the reverse leakage of the bipolar transistors and the common-emitter current gain (*β*) dominated by highly doped silicon remained unchanged with the decrease in the neutron flux. Our work demonstrates that the ELNFS effect in boron-doped silicon can be well explained by noise diagnostic analysis together with electrical methods and simulation, which thus provide the basis for detecting the enhanced low-particle-flux damage effect in other semiconductor materials.

## 1. Introduction

The outer space radiation induced by high-energy particles such as electrons, protons, neutrons, γ-rays and heavy ions can cause radiation-related damages and even the malfunction or even failure of the spacecraft electronic system. With the rapid development of space technologies, the semiconductor device performance and underlying physics of electronic components under radiation are widely concerned and have been extensively explored, and various space radiation effect models have been proposed. Total ionizing dose (TID) [1,2] and enhanced low dose rate sensitivity (ELDRS) effects [3,4] are the main causes for the functional degradation and performance failure of semiconductor devices.

The TID and ELDRS effects are common and well-known, e.g., in space-based applications of semiconductor devices. The TID and the ELDRS effects are induced by ionization damages in semiconductor devices. The TID effect can greatly shorten the service life of space communication systems, and the ELDRS effect can accelerate the TID effect. In addition, the ELDRS effect can greatly reduce the damage threshold of semiconductor devices because the radiation can destroy semiconductor devices under a low absorption dose rate [4]. Even the small-dose radiation can cause the performance degradation of semiconductor devices and the failure of the whole electronic system, which is a disaster to a spacecraft. Therefore, the ELDRS effect in semiconductor devices should be further explored.

A spacecraft electronic system is threatened by the irradiation of neutrons, electrons, protons, γ-rays, heavy particles and cosmic rays in the outer space [5,6,7,8], which can cause ionization and displacement damages in semiconductor devices. Ionization damage leads to the ELDRS effect in semiconductor devices, but the induction of ELPFS effect by displacement damage is still unknown. In 1991, based on deep level transient spectroscopy, Hallèn et al. [9] found that under 1.3 MeV protons with fluence of 5 × 10^9^ cm^−2^, as the flux decreased from 2.37 × 10^10^ cm^−2^ s^−1^ to 1.25 × 10^7^ cm^−2^ s^−1^, the stable vacancy-related point defect density in silicon increased. In 1993, B. G. Svensson et al. found that the defect densities in the silicon (100) crystals produced by 1.0 MeV ^76^Ge and 2.3 MeV ^120^Sn decreased with the increase in the ion flux [10]. The ELPFS phenomena are similar to ELDRS phenomena and can be interpreted as the ELPFS effect. However, these effects are seldomly investigated, and the physical mechanism remains unknown.

Neutrons, electrons, protons and heavy particles in space can interact with semiconductors and generate vacancies and interstitials inside the materials. Meanwhile, the flux of the above particles also changes with time and latitude. When traveling in space, these semiconductor devices of spacecraft electronic systems might be increasingly damaged, shortening their lifetime due to the low flux of particles.

Messenger et al. [11] believed that the displacement damage of semiconductor devices is mainly ascribed to the displacement damage of silicon. Therefore, the ELPFS effect in semiconductor devices can be understood by studying the ELPFS effect in silicon materials. In this study, with the neutron pulse reactor as the irradiation source, the ELNFS effect in boron-doped silicon with different doping concentrations was studied by the noise diagnostic analysis and the reason for ELNFS effect in boron-doped silicon was explored by the simulation based on the theory of diffusion-limited reactions. Furthermore, the relationship between the ELNFS effect and the doping concentration was also studied based on the theory of diffusion-limited reactions and noise diagnostic analysis. The ELNFS effect in p^+^-n-p Si-based bipolar transistors introduced by boron-doped silicon with different doping concentrations was explored to verify the results of the noise diagnostic analysis and simulation.

## 2. Materials and Methods

### 2.1. Irradiation Experiments and Noise Tests of Silicon

The highly doped p-type and lowly doped p-type silicon (110) single crystals were supplied by Hefei Kejing Materials Technology Co., Ltd. (Hefei, Anhui, China). The indicators of the materials are provided in Table 1.

The neutron irradiation environment is provided by CFBR-II reactor, which is operated by Institute of Nuclear Physics and Chemistry in China’s Academy of Engineering Physics. It is a fast-burst fission reactor able to offer both transient and steady neutron irradiation environments. The spectrum of CFBR-II reactor is analogous to fission spectrum, and the average neutron energy is 1.2 MeV. Calibrated boron ionization chambers, whose current reply is linear to neutron flux in the range concerned, are placed at fixed positions to monitor the power of reactor. The total neutron fluence was 7 × 10^13^ n cm^−2^ and the neutron fluxes applied in the experiments were 5 × 10^9^ n cm^−2^ s^−1^, 2 × 10^10^ n cm^−2^ s^−1^, 4 × 10^10^ n cm^−2^ s^−1^ and 5 × 10^10^ n cm^−2^ s^−1^, respectively. Ten parallel samples of boron-doped silicon were selected for each flux, and the sample size was 1 mm × 1 mm × 0.5 cm. The noise data of the samples before and after irradiation were tested and selected by the noise parameter testing system. The above neutron irradiation and noise tests were all carried out at room temperature.

The noise spectra of boron-doped silicon samples were measured by the noise test system [12]. During the noise acquisition process, each sample was positioned in the middle of the adapters, which can be automatically levelled on the sample surface under the control of a spring, so that the metal electrode could apply the voltage on both sides of the sample. In order to avoid the interference of the local electrical frequency, the test was carried out in an electromagnetism shielding box located in the EM shielded room. The noise parameter collection system was fixed on the high-precision mechanically damped platform to reduce the vibration impact on the system. The noise parameters of semiconductor materials and devices are stationary random variables with ergodicity, indicating that the probability that all possible values of noise of semiconductor materials occur at different times over a long period is equal to the probability that all possible values occur at fixed time. In the noise calculation process, the statistical average value was used to approximate the real value. In the noise measurement process, the measurement data were collected for several times and averaged to reduce the statistical error. In this study, the noise power spectral density of each sample was the average value of 10^5^ measurement data.

The noise of semiconductor is usually composed of white noise (shot noise and thermal noise), 1/f noise and generation-recombination noise (G-R noise). The noise power spectral density can be defined as follows:(1)$$S\left(f\right)=A+\frac{B}{f^{\gamma }}+\frac{C}{1+{\left(f/f_{0}\right)}^{\alpha }},$$
where *A* is the white noise; the second term is 1/f noise; the third term is G-R noise; *B* is the amplitude of the 1/f noise; *C* is the amplitude of the G-R noise; *f*_0_ is the corner frequency of the G-R noise; the indices *γ* and *α* are usually equal to 1.

Among all the three types of noise, though the white noise exists in the whole spectral range, the noise power spectral density is relatively small and can be easily drowned by 1/f noise and G-R noise in the low-frequency bands. Both 1/f noise and G-R noise can be displayed in the noise power spectra, but only the noise with large enough magnitude can be displayed in the noise power spectra. In addition, the noise can be experimentally observed when there is a magnitude difference between the two kinds of noises. If there is no generation source of 1/f noise or G-R noise in semiconductor materials, the 1/f noise or G-R noise cannot be experimentally observed.

### 2.2. Irradiation Experiment and Electrical Test Information of Si-Based Bipolar Transistors

In a p^+^-n-p Si-based bipolar transistor, the lowly doped silicon is the collector and the highly doped silicon is the emitter. Therefore, a p^+^-n-p Si-based bipolar transistor can be used to study the neutron flux sensitivity effect in the semiconductor devices. The function of the highly doped silicon and lowly doped silicon in bipolar transistors can be characterized by the common-emitter current gain (*β*) and the collector-emitter leakage current (*I_CBO_*) between the collector and the base with the emitter-base junction open, respectively.

Figure 1 shows various current components in a p^+^-n-p Si-based bipolar transistor under a common emitter configuration. In the active mode, the emitter-base junction is forward-biased (*U_BB_* > 0), and the base-collector junction is reverse-biased (*U_CC_* < 0). The holes injected from the emitter constitute the current *I*_EP_, which is the largest current component in a well-designed transistor. Most of the injected holes reach the collector junction and give rise to the current *I_CP_*. There are three base current components, labeled *I_BP_*, *I_EN_*, and *I_CBO_*, respectively. *I_BP_* corresponds to the electrons supplied by the base and required to replace the electrons recombined with injected holes (i.e., *I_BP_* = *I_EP_* − *I_CP_*). *I_EN_* corresponds to the current arising from the electrons injected from the base to the emitter. *I_CBO_* corresponds to thermally generated electrons that are near the base-collector junction edge and drift from the collector to the base.

In terms of the various current components, the terminal currents are described as below:(2)$$I_{E}=I_{EP}+I_{EN},$$
(3)$$I_{C}=I_{CP}+I_{CBO},$$
(4)$$I_{B}=I_{EN}+I_{BP}-I_{CBO},$$

The *β* is the incremental change in *I_C_* with respect to an incremental change in *I_B_*. As for a common emitter configuration, *I_E_* = *I_B_* + *I_C_*, so the *β* can be expressed by:(5)$$\beta =\frac{I_{CP}}{I_{E}-I_{CP}}=\frac{I_{CP}}{I_{EN}+I_{BP}}=\frac{I_{C}-I_{CBO}}{I_{B}+I_{CBO}}.$$

The neutron radiation damage in lowly doped p-type silicon leads to the increase in *I_CBO_* of transistor. If the lowly doped p-type silicon is sensitive to the ELNFS effect, the *I_CBO_* increases with the decrease in the flux. *I_C_* and *I_B_* are generally much larger than *I_CBO_*, so *β* is mainly affected by the highly doped p-type silicon in the emission according to Equation (5) and can be used to characterize the ELNFS effect in the highly doped p-type silicon.

In this study, PNP bipolar transistors 3CK4B were used as the test samples. The transistors 3CK4B are p^+^-n-p Si-based bipolar transistor that take the lowly boron-doped silicon as the collector and the highly boron-doped silicon as the emitter. The doping concentrations of lowly doped and highly doped silicon in the transistors 3CK4B are of the same magnitude as that of the silicon in Table 1. The *β* and *I_CBO_* of transistors 3CK4B pre- and post-irradiation were measured by Keithley 4200A-SCS Parameter Analyzer (Beaverton, OR, USA). The transistors were irradiated until the fluence reached 7 × 10^13^ n cm^−2^ at different fluxes. The neutron fluxes in the experiments were set as 5 × 10^9^ n cm^−2^ s^−1^, 2 × 10^10^ n cm^−2^ s^−1^, 4 ×10^10^ n cm^−2^ s^−1^, and 5 × 10^10^ n cm^−2^ s^−1^, respectively.

## 3. Results and Discussion

### 3.1. Relationship between the Voltage Noise Power Spectral Density of Boron-Doped Silicon with Different Doping Concentrations and Neutron Flux

Low frequency noise, especially 1/f and G-R noise, can be used to sensitively characterize internal defects in electronic materials and devices. It has already been used to characterize radiation damage in them. Due to radiation, the internal defects and structural damage in electronic materials and devices increases, leading to an increase in low frequency noise power spectral density [13,14]. In 1995, Babcock et al. studied the radiation resistance of UHV/CVD SiGe HBT irradiated by a ^60^Co γ-ray, and found that an increase in the total absorbed dose resulted in performance degradation in the SiGe HBT, with the current gain *β* decreasing with increasing absorbed dose, and a reduction in internal defects [15].

We explore the distribution uniformity and types of internal defects in boron-doped silicon by low frequency voltage noise power spectral density (*S_V_*) and analyze the number of defects in boron-doped silicon by amplitude of low frequency noise power spectral density.

The curves 1–40 in Figure 2 show the *S_V_* of lowly doped p-type silicon at different fluxes. In the frequency range of 1–10^4^ Hz, the noise spectra of lowly doped p-type silicon with different fluxes show both 1/f and white noise. The amplitude of white noise *A* is approximately equal to 2.35 × 10^−16^ V^2^ Hz^−1^. In the frequency range of 1–10^3^ Hz, the noise response of lowly doped p-type silicon is of 1/f noise type:(6)$$S\left(f\right)=\frac{B}{f^{\gamma }},$$

Within the neutron flux range of 5 × 10^9^–5 × 10^10^ n cm^−2^ s^−1^, *γ* is close to 1 for all the lowly doped p-type silicon. However, the *S_V_* curves of all the 10 parallel samples varies with the flux, thus resulting in scattered data points in the amplitude of 1/f noise (*B*). When the flux is low, there is greater discreteness in *B*. The difference between the parallel samples is due to the uneven distribution of impurities in the silicon. The physical reason why the curves within each batch differ is explained later.

The curves 1–40 in Figure 3 show the *S_V_* of highly doped p-type silicon at different fluxes. The curves in Figure 2 and Figure 3 can reflect the influence of doping concentration and defect distribution on *S_V_*. In the frequency range of 1–10^5^ Hz, the noise response under different fluxes is a combination of 1/f noise and G-R noise, while white noise is completely suppressed.

These results indicate that within the neutron flux range of 5 × 10^9^–5 × 10^10^ n cm^−2^ s^−1^, the *γ* value of all the highly doped p-type silicon is approximately equal to 1, and the *B* varies in the range of 5.15 × 10^−11^–3.72 × 10^−8^ V^2^ Hz^−1^. Under different fluxes, the *S_V_* curves of samples show similar variations. As the flux decreased, the data variation in the *S_V_* curves of parallel samples is increased. The amplitude of G-R noise *C* shows greater variation within the *C* range of 5.82 × 10^−14^–0.65 × 10^−12^ V^2^ Hz^−1^. Therefore, the corner frequency of G-R noise (*f*_0_) is directly correlated with the energy levels and types of defects [16]. G-R noise with specific defects reflects the specific corner frequency, and the defect number can be directly correlated with the G-R noise amplitude. Therefore, the change in defect type might result in the offset of corner frequency, and the change in the defect number might lead to variations of the G-R noise amplitude. The amplitude of G-R noise of highly doped p-type silicon varies, but the corner frequency remains steady and is approximately equal to 200 Hz, indicating the existence of a large amount of defects inside materials. In addition, these defects are related to boron atom acceptor. Compared with lowly doped p-type silicon, the highly doped samples exhibit more defects to yield observable G-R noise.

In this work, *B* is used to characterize the ELNFS effect in boron-doped silicon. Figure 4a,b shows the comparison of *B* in the lowly and highly doped p-type silicon under different neutron fluxes. Within the neutron flux range of 5 × 10^9^–5 × 10^10^ n cm^−2^ s^−1^, the lowly doped p-type silicon is strongly sensitive to ELNFS effect, and the *B* increased as the neutron flux decreased. Shown as the red fitting line in Figure 4a, the relationship between *B* and the neutron flux (*φ*) is expressed as:(7)$$B\propto \varphi^{-2.1},$$

The *S_V_* − *f* curves of post-irradiated boron-doped silicon also demonstrate 1/f noise feature. Based on the Hooge formula, the relational expression between *S_V_* and carrier number of the samples (*N*) is
(8)$$S_{V}\left(f\right)=\frac{B}{f^{\gamma }}\propto \frac{\alpha_{H}}{Nf},$$
where *α_H_* is Hooge coefficient; *f* is frequency. Due to the carrier removal effect of defects [17,18], the decrease in *N* results in the increase in *S_V_*. Compared with a high neutron flux, the low flux can generate more defects, which enhance the carrier removal efficiency. Therefore, the *B* increases with the increase in *S_V_* under a low neutron flux. The above changing trend is responsible for the observed ELNFS effect in the boron-doped silicon with different doping concentrations in this study.

The *B* of highly doped p-type silicon does not increase as the neutron flux decreased, indicating that the highly doped p-type silicon is not sensitive to ELNFS. When the neutron flux is 5 × 10^9^ n cm^−2^ s^−1^, the variation of the *B* was the largest. The increase in the variation of *B* by a low neutron flux might be interpreted as follows. The preparation process of boron-doped silicon introduced a variety of impurities [19,20,21] including donors and acceptors, which are not distributed uniformly inside the lattice. When neutron interacts with silicon with different types and amounts of impurities, the types and quantity of generated defects [19,20] are significantly different. However, the differences in the quantity and type of defects remain when the neutron flux is low. Therefore, the removal effect for the defect carrier in every sample is different, and the carrier quantity in every sample is also different. According to Equation (8), the low neutron flux leads to the relatively large discreteness.

In this study, only the lowly doped p-type silicon was highly sensitive to ELNFS effect, indicating that ELNFS effect might be related to a low doping concentration. Pease et al. [17] studied the 1 MeV neutron carrier removal rate of the n- and p-type dopants in silicon with the doping concentration of 2 × 10^14^ cm^−3^–10^15^ cm^−3^. It was demonstrated that under neutron irradiation, the carrier concentration (*n*) in Si-based MOSFETs was related to the neutron fluence (*Φ*):(9)$$n=n_{0}-k\times \Phi ,$$
where *n*_0_ is the carrier concentration before irradiation; *k* is the carrier removal rate. Stein et al. [22] found that *k* varied between 4 cm^−1^ and 5 cm^−1^ for p-type silicon [17]. In this study, the *k* value is set as 4 cm^−1^ for p-type silicon. ∆*S_V_* is the noise power spectral density change under neutron irradiation and can be calculated by Equation (8) as follows:(10)$$\Delta S_{V}=S_{V}\left(N_{0}\right)-S_{V}\left(N\right)\propto \frac{\alpha_{H}}{N_{0}}-\frac{\alpha_{H}}{N}=\frac{\alpha_{H}\left(N-N_{0}\right)}{NN_{0}},$$
where *S_V_* (*N*_0_) is the noise power spectral density before irradiation; *S_V_* (*N*) is the noise power spectral density after irradiation. Therefore, the change rate of *S_V_* induced by irradiation (Δ*S_V_/S_V_*_0_) is expressed as:(11)$$\Delta S_{V}/\Delta S_{V0}\propto \frac{N-N_{0}}{N},$$
where *W* is the material volume. The carrier concentration of boron-doped silicon irradiated by neutron can be calculated by Equation (9). *N* can be calculated by Equation (12):(12)N=nW.

Substituting Equations (9) and (12) into Equation (11) yields
(13)$$\Delta S_{V}/\Delta S_{V0}\propto \frac{N-N_{0}}{N}=\frac{-k\times \Phi }{n_{0}-k\times \Phi }.$$

In Table 2, the Δ*S_V_/S_V_*_0_ within the neutron fluence range from 1 × 10^13^ n cm^−2^ to 1 × 10^15^ n cm^−2^ is calculated by Equation (13). For the lowly doped p-type silicon, the Δ*S_V_/S_V_*_0_ is approximately equal to 1, suggesting the Δ*S_V_* induced by neutron irradiation is enough to change the *S_V_* greatly, suggesting ELNFS effect is obvious. However, the Δ*S_V_/S_V_*_0_ range of highly doped p-type silicon is 4.00 × 10^−6^–4.00 × 10^−4^ and the variation is so small that it is not enough to change *S_V_* greatly. The Δ*S_V_/S_V_*_0_ of highly doped p-type silicon might be concealed by test errors, demonstrating ELNFS effect is not obvious.

### 3.2. Mechanism of ELNFS in p-Type Silicon

Neutron flux effects in silicon are related to the competition between the formation of stable vacancy-associated defects, and the recombination of vacancies, interstitials, or remaining vacancy-related defects resulted in the more severe damage [23]. To analyze the experimental results, the same simulation model is used [23]. In this model, the defect evolution is simulated based on the theory of diffusion-limited reactions [24,25], and a sequential build-up of interstitial, vacancy and complex defects with diffusion/dissociation reactions are assumed as core ingredients. The rate of agglomeration reaction is given by 4πR (*D_A_ + D_B_*) (*A*)(*B*), where *D_A_* and *D_B_* represent the diffusion constants. (*A*) and (*B*) indicate the concentrations of the two species A and B, respectively. The rate of dissociation reaction can also be given by *C_diss_* (*A*) = *V_diss_* EXP (−E_diss_/*k_B_T*) (*A*), where *V*_diss_ is the vibrational frequency of dissociation for a particular species; *E_diss_* is the energy associated with that dissociation; *k_B_* is Boltzmann constant; *T* is the temperature. In the p-type samples, the dopant is boron. The most important reaction related to boron is the reaction between boron interstitial and oxygen interstitial to form a stable defect [4]. Hence, all the factors and defect reactions chosen for the defect diffusion-recombination model are the same as those in the previous report except that the impurity is set to be oxygen because there was no phosphorus in irradiated samples [25]. The initial concentration of oxygen interstitials ((*I_O_*)) has been set as 2 × 10^17^ cm^−3^ [4]. Since boron could react with oxygen, the higher concentration of boron decreased oxygen interstitials. The simulation results under different concentrations of oxygen interstitials are shown in Figure 5. (*V*_r_) is the concentration of remaining vacancy-related defects.

When the concentration of oxygen interstitials is in the range of 2 × 10^15^ cm^−3^ to 2 × 10^17^ cm^−3^ and the neutron fluence is constant, the remaining vacancy-related defects under low flux are more than those under high flux due to ELNFS effect. As oxygen interstitials concentration decreased, the ratio of the remaining vacancy-related defects under low flux to those under high flux also decreased, indicating that ELNFS effect could be attenuated by decreasing the concentration of oxygen interstitials. Under the same concentration of oxygen interstitials, the concentrations of remaining vacancy-related defects in respect to neutron fluence at four neutron fluxes are simulated based on the theory of diffusion-limited reactions (Figure 6a). In Figure 6a, the data points are simulated based on the theory of diffusion-limited reactions. The lines are fitting lines of data points. (*V*_r_) increases with the increase in neutron fluence and the decrease in neutron flux. (*V*_r_) under the neutron flux is 5 × 10^9^ n cm^−2^ s^−1^, which is 1.3 times higher than that under 5 × 10^10^ n cm^−2^ s^−1^. Under the neutron fluence of 7 × 10^13^ n cm^−2^, (*V*_r_) increases with the decrease in the neutron flux (Figure 6b). The relationship between the concentration of remaining vacancy-related defects and *φ* is expressed as follows:(14)$$\left(V_{r}\right)\propto \varphi^{-0.12},$$

Equations (7) and (14) suggest that the *B* and the (*V*_r_) in lowly doped p-type silicon are inversely proportional to the neutron flux, as shown in Figure 6b. Therefore, we propose that ELNFS in p-type silicon might be caused by the difference in the number of remaining vacancy-related defects under different neutron fluxes.

As shown in Figure 5, the number of remaining vacancy-related defects is on the order of 1 × 10^15^ cm^−3^ under different oxygen interstitials. In the samples with a doping concentration of 1 × 10^13^ cm^−3^, the carriers are mainly intrinsic carriers and vacancy-related defects, whose concentration is on the order of 1 × 10^15^ cm^−3^, and significantly affected the electrical properties of the samples. In the samples with a doping concentration of 1 × 10^19^ cm^−3^, whose carriers are mainly from acceptor impurities, the concentration of carriers was close to 1 × 10^19^ cm^−3^. However, (*V*_r_) remains to be in the order of 1 × 10^15^ cm^−3^, which is four orders of magnitude smaller than that of carriers. Therefore, the influence of neutron flux is the least in highly doped samples.

The ELNFS effect in p-type samples with different doping concentrations might be interpreted from the following two aspects. Firstly, with the increase in the doped boron concentration, the oxygen interstitials concentration is decreased, thus suppressing the neutron flux effect. Secondly, the difference in the number of remaining vacancy-related defects caused by different neutron fluxes is sensitive to the doping concentration. As the doping concentration increased, the influence of the neutron flux is also suppressed gradually. The above analysis is also consistent with the theoretical calculation results of noise in Table 2.

### 3.3. ELNFS Effect in the Si-Based Bipolar Transistors

The lowly and highly doped p-type silicon discussed above are used as both the substrates and the main functional layers in semiconductor devices. For example, the lowly doped p-type silicon is often used as the collector in bipolar transistors. Therefore, the radiation resistance of these materials can greatly affect the radiation resistance of semiconductor devices. When the Si-based devices are exposed to irradiation, the rapid damage of the material can increase the dark current and deteriorate reverse pressure resistance characteristics. In addition, the charge collection ability of the collector will also be impaired, thus decreasing the gain of bipolar transistors.

The above results of noise characterization and defect simulation demonstrate existence of ELNFS effect in boron-doped silicon materials. In addition, ELNFS effect could be greatly reduced by increasing the doping concentration. Furthermore, we also study the influence of the ELNFS effect on the performance of Si-based bipolar transistors induced by boron-doped silicon materials. The *β* and *I_CBO_* are used to characterize ELNFS effect in Si-based bipolar transistors induced by the lowly and highly doped p-type silicon, respectively. The *I_CBO_* of Si-based bipolar transistors is increased with the decrease of neutron flux (Figure 7a), and the *I_CBO_* is equal to 4 × 10^−10^ A under the neutron flux of 5 × 10^9^ n cm^−2^ s^−1^, which is one order of magnitude higher than that under other three neutron fluxes, indicating ELNFS effect in the lowly doped silicon increases the reverse leakage of bipolar transistors.

For the bipolar transistor, the theoretical expectation of *I_CBO_* after neutron irradiation is [26,27]:(15)$$I_{CBO}=-\frac{Sqn_{i}W_{D}}{2\tau }$$
where *S* is the area of Collector–Base junction; *q* is the electron charge; *n_i_* is the intrinsic carrier concentration; *W_D_* is the length of depletion region; *τ* is the hole lifetime.

Under reverse bias, the *W_D_* of Base-Collector junction can be calculated by:(16)$$W_{D}=\sqrt{\frac{2V_{a}\varepsilon_{S}}{qN_{D}}}$$
where *ε_s_* is dielectric constant; *V_a_* is applied voltage; *N_D_* is collector doping concentration.

The displacement damage (DD) introduces bulk traps in the semiconductor, which can lead to a significant reduction in lifetime. The change of the reciprocal of the minority carrier lifetime (1/Δ*τ*) increases linearly with the increase of the DD-induced bulk traps [26,27], that is,
(17)$$\frac{1}{\Delta \tau }=\frac{1}{\tau_{post}}-\frac{1}{\tau_{pre}}=\delta v_{t}\Delta \left[V_{r}\right]V_{O}$$
where *τ_pre_* and *τ_post_* are hole lifetimes before and after neutron irradiation; *δ* is the capture cross section for carriers; *v_t_* is the thermal velocity, and *V_O_* is the volume of collector. Δ(*V*_r_) can be simulated based on the theory of diffusion-limited reactions.

Substituting (16) and (17) into (15), we obtain the equation of the theoretical expectation of *I_CBO_*, which takes the form
(18)$$I_{CBO}=-\frac{Sqn_{i}}{2}\sqrt{\frac{2V_{a}\varepsilon_{S}}{qN_{D}}}\times \left(\delta v_{t}\Delta \left(\left[V_{r}\right]V_{O}\right)+\frac{1}{\tau_{pre}}\right)$$

According to Equation (18), the theoretical expectation of *I_CBO_* is obtained. In Figure 7a, the blue data points are the theoretical expectation of *I_CBO_* under four neutron fluxes, and the blue line is the theoretical expectation of *I_CBO_*.

It can be seen from Figure 7a that both the experiment and the theoretical expectation of *I_CBO_* increase nonlinearly with the decrease in neutron flux, but they are not completely consistent. We guess that the difference between experiment and theoretical expectation is due to the oversimplification of theoretical expectation, which fails to fully consider the process structure inside the transistors.

When the *U_BB_* ranged from −0.85 V to −0.3 V and *U_cc_* is −1 V, the *β* of Si-based bipolar transistors remains unchanged as the neutron flux decreased (Figure 7b), indicating ELNFS effect in boron-doped silicon could be greatly reduced by increasing doping concentration.

Bipolar devices may be sensitive to ELNFS effect due to introduction of lowly doped silicon as electrodes. Therefore, it is necessary to consider the enhanced damage in low-neutron-flux environment. Within a certain energy range, the removal ability of protons is equivalent to that of neutrons for the carriers in semiconductor devices, and in certain circumstances, the two kinds of particles can be replaced with each other [17,28,29]. Therefore, the neutron source can be used to simulate high-energy protons in space. This work demonstrates that the low-proton-flux irradiation damage enhancement may exist in silicon. In the space environment, protons are one of the main sources of radiation, which can lead to the single event effect and the functional turbulence of the spacecraft electronic system. It is important to study whether or not the enhanced low-proton-flux sensitivity effect in semiconductor materials can enhance the single event effect in the electronic system. High doping concentration allows the better performances including defect tolerance. Therefore, in the neutron or high-energy proton radiation environment, the semiconductor devices that epitaxially grow on highly doped silicon should be selected as the components of electronic systems in order to minimize the ELPFS effect.

## 4. Conclusions

This work reports the ELNFS effect in boron-doped silicon. Within the neutron flux range of 5 × 10^9^–5 × 10^10^ n cm^−2^ s^−1^, lowly doped p-type silicon is highly sensitive to ELNFS effect, and the *B* increases as the neutron flux decreases. Noise diagnostic analysis and simulation based on the theory of diffusion-limited reactions indicate that ELNFS effect in boron-doped silicon become weak as doping concentration increased. The weakened ELNFS effect might be interpreted as follows. With increasing doping concentration than (*V*_r_), the effect of the remaining vacancy-related defects on the material becomes weak. The ELNFS effect becomes negligible when the doping concentration is 10^19^ cm^−3^. The Δ*S_V_/S_V_*_0_ range for highly doped p-type silicon is 4.00 × 10^−6^ to 4.00 × 10^−4^ when the neutron fluence range is from 1 × 10^13^ n cm^−2^ to 1 × 10^15^ n cm^−2^. However, the ELNFS effect in lowly doped p-type silicon is obvious. Therefore, Δ*S_V_/S_V_*_0_ is approximately equal to 1. When lowly doped p-type silicon is widely used as collector electrode in semiconductor devices, the reverse leakage current of the semiconductor devices increases. Furthermore, the noise diagnostic analysis and simulation can well interpret the ELNFS effect in silicon and provide the basis for detecting the enhanced low-particle-flux damage effect in other semiconductor materials.

## Figures and Tables

**Figure 1 nanomaterials-10-00886-f001:**
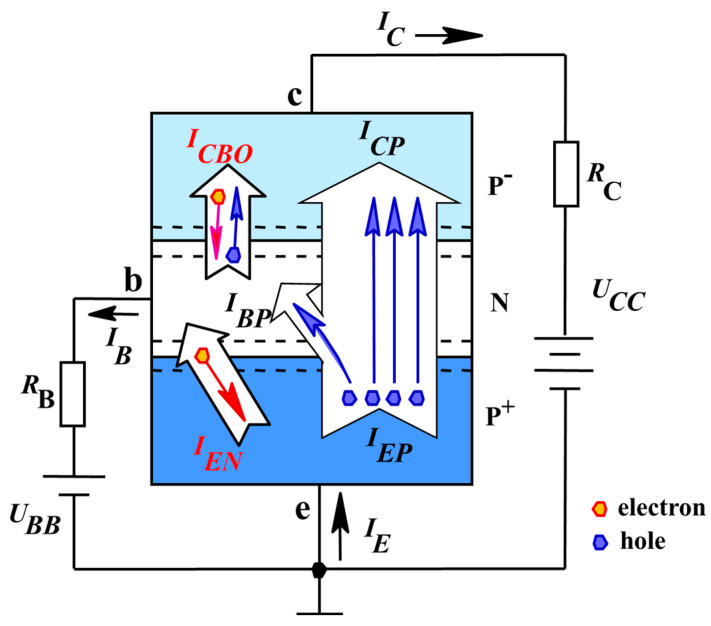
Various current components in a p^+^-n-p Si-based bipolar transistor under a common emitter configuration.

**Figure 2 nanomaterials-10-00886-f002:**
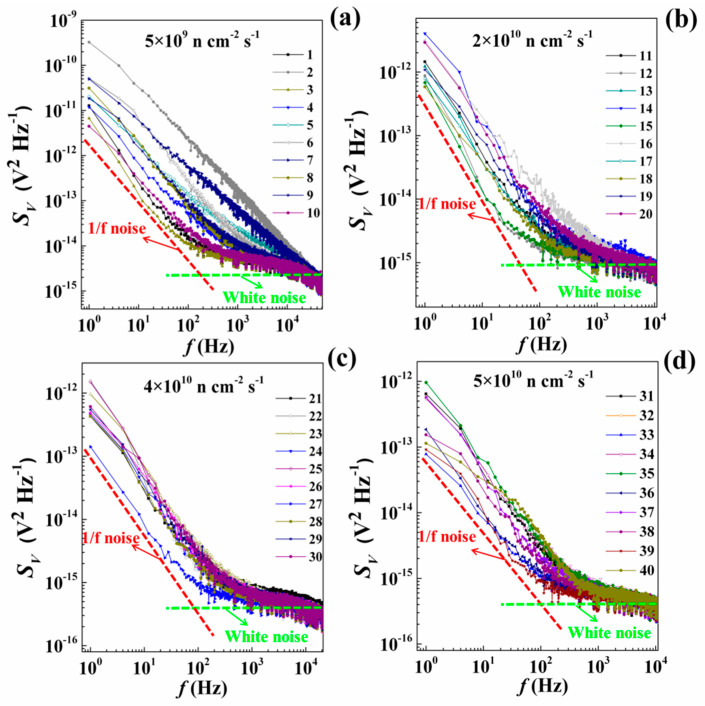
*S_V_* curves for lowly doped p-type silicon at different fluxes, (**a**) when flux is 5 × 10^9^ n cm^−2^ s^−1^; (**b**) when flux is 2 × 10^10^ n cm^−2^ s^−1^; (**c**) when flux is 4 × 10^10^ n cm^−2^ s^−1^; (**d**) when flux is 5 × 10^10^ n cm^−2^ s^−1^.

**Figure 3 nanomaterials-10-00886-f003:**
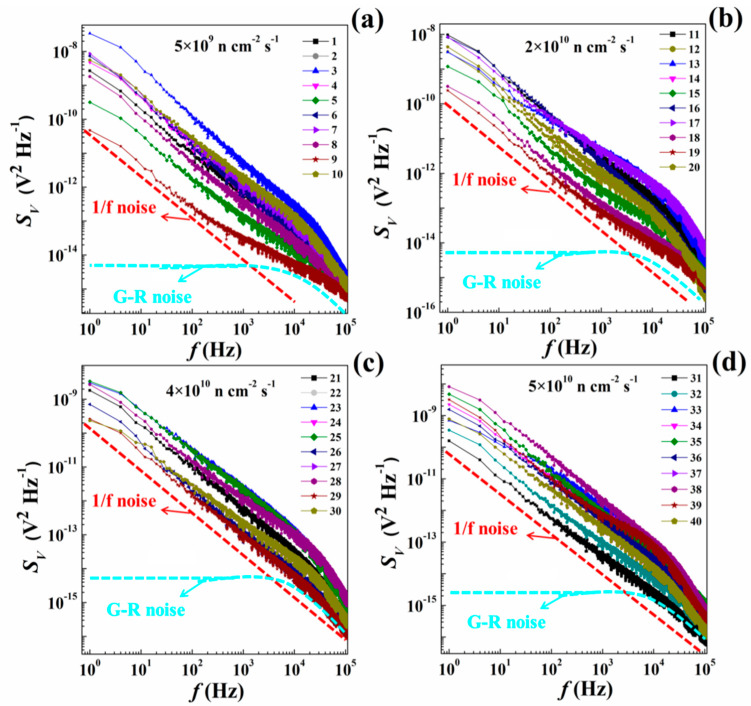
*S_V_* curves for highly doped p-type silicon at different fluxes, (**a**) when flux is 5 × 10^9^ n cm^−2^ s^−1^; (**b**) when flux is 2 × 10^10^ n cm^−2^ s^−1^; (**c**) when flux is 4 × 10^10^ n cm^−2^ s^−1^; (**d**) when flux is 5 × 10^10^ n cm^−2^ s^−1^.

**Figure 4 nanomaterials-10-00886-f004:**
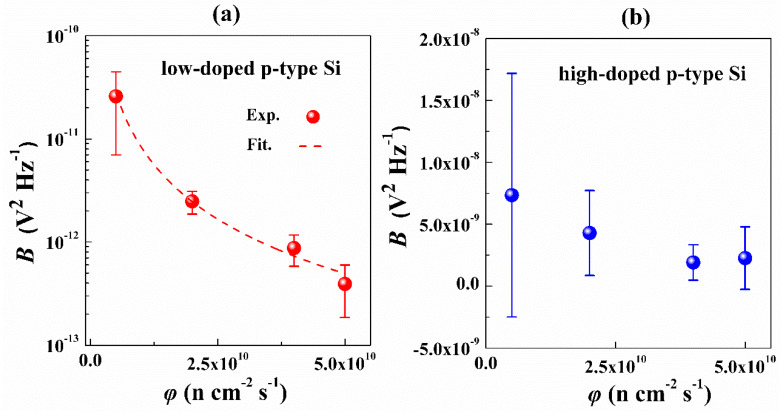
Relationship between the *B* of boron-doped silicon and the *φ*, (**a**) when Boron doping concentration is 10^13^ cm^−3^; (**b**) when Boron doping concentration is 10^19^ cm^−3^.

**Figure 5 nanomaterials-10-00886-f005:**
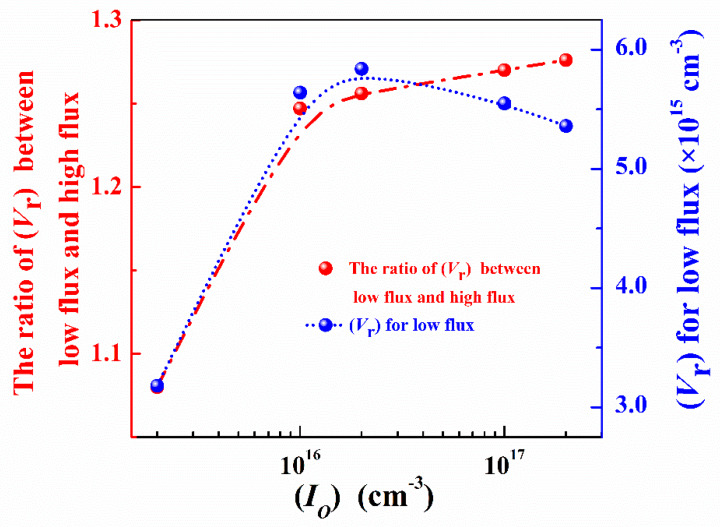
Dependencies of the ratio of remaining vacancy-related defects under a low flux to those under a high flux and concentrations of remaining vacancy-related defects under a low flux on concentrations of oxygen interstitials.

**Figure 6 nanomaterials-10-00886-f006:**
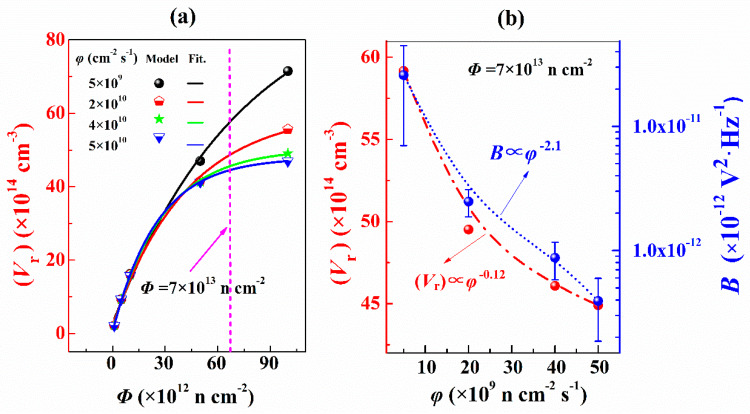
(**a**) (*V*_r_) against neutron fluence; (**b**) (*V*_r_) and the *B* against neutron flux under the neutron fluence of 7 × 10^13^ n cm^−2^.

**Figure 7 nanomaterials-10-00886-f007:**
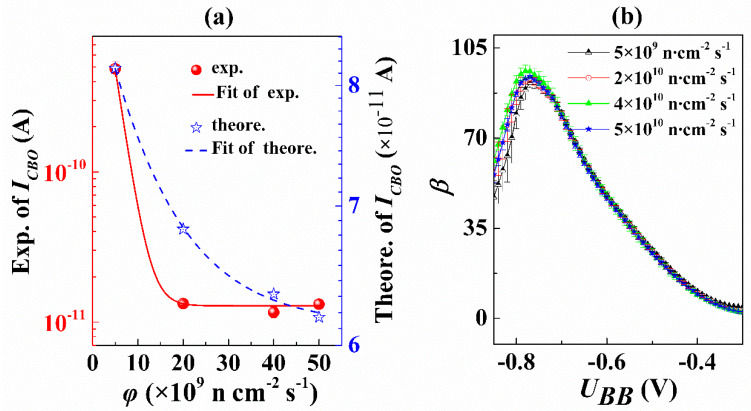
(**a**) *I_CBO_* against neutron flux; (**b**) *β* against *U_BB_* at four neutron fluxes and the neutron fluence of 7 × 10^13^ n cm^−2^.

**Table 1 nanomaterials-10-00886-t001:** Silicon used in the study.

No.	Materials	Crystal Orientations	Descriptions
1	highly doped p-type silicon	(110)	Doped element: Boron Doping concentration: 10^19^ cm^−3^
2	lowly doped p-type silicon	(110)	Doped element: Boron Doping concentration: 10^13^ cm^−3^

**Table 2 nanomaterials-10-00886-t002:** Δ*S_V_/S_V_*_0_ under different neutron fluences.

*Φ* n·cm^−2^	Δ*S_V_/S_V_*_0_ Lowly Doped p-Type Si	Δ*S_V_/S_V_*_0_ Highly Doped p-Type Si
1 × 10^13^	1.33	−4.00 × 10^−6^
7 × 10^13^	1.04	−2.80 × 10^−5^
1 × 10^14^	1.03	−4.00 × 10^−5^
1 × 10^15^	1.00	−4.00 × 10^−4^

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
