# Peer review of "Enhanced Low-Neutron-Flux Sensitivity Effect in Boron-Doped Silicon"

_nanomaterials, 2020, doi:10.3390/nano10050886_

Round 1

Reviewer 1 Report

The paper "Enhanced Low-Neutron-Flux sensitivity Effect in Single Crystal Silicon" deals with the neutron-induced irradation damage effects in semiconductors.
These are of high relevance e.g. in harsh environments like on spacecrafts.
The paper adds new insight to the mechanism of low-neutron-flux sensitivity effects by analyzing the electronic noise behaviour.

Overall, the paper is well-intended, adresses relevant questions and provides experimental data in combination with theoretical interpretation.
Generally, I think the scope of the paper is more on microelectronics than on nanomaterials.
Thus I think it should better be transferred to the respective MDPI journal, as I think the subject is outside the scope of MDPI Nanomaterials.
Moreover, there are a few things to be improved:
While the title and also some of the formulations (e.g. the first sentence of the conclusionn) lead to the assumption that pure crystal silicon is regarded,
it is in fact dealing with boron-doped Si (with different concentrations). This should be pointed out e.g. by changing the title of the paper and the relevant formulations.

The description of the experimental method is rather short with regard to the fact that using of a neutron pulse reactor is by far not a standard technique. More details e.g. regarding the neutron energy and its distribution should be given, or at least some accessible reference been cited. It should also be introduced more accentuated and not just by a subclause in l. 75. By this it can also be avoided to repeat the mention in the beginning o 2.1 and the end of 2.2

The authors should use a different term than "simulation" for their measurements. Although it is understood that the simulation of e.g. irradiation in space is meant, this might be misleading with regard to a purely numerical study.

In 2.2 it should be explained if there is a certain reason for chosing the 3CK4B transistor.

Some specific comments:

l. 49 : its better to write "The TID and ELDRS effects are common and long time well-known e.g. in space-based applications of electronisc devices."
l. 73 : for clearence, e.g. "semiconductor" should be added before "devices".
l. 96 : what "adapters" are meant ?
l. 101: I think the resistance against eathquakes is someehow irrelevant in this context, "mechanically damped" or "stabilized" or similar would be suffficient.
Fig. 2 and 3: what is the meaning of the different curves 1 - 40 ?
Is this just a sample spread which can be averaged, or is there a physical reason why the curves within each batch differ.
These figs are rather voluminous and, if possible, should be reduced so a more compact, summarizing presentation with less details.

l. 216 : I would sggest to replace inside the math formula "flux" by some defined math symbol, e.g. φ
Figs. 2,4a,5,6,7a: The drawn lines in some cases obviously are fit funcctions (6b), in others just lines or splines (?).
In these cases they should be omitted or at least explained and justified (to guide the eye).

Reviewer 2 Report

Manuscript reference number: nanomaterials-767470
Title: Enhanced Low-Neutron-Flux Sensitivity Effect in Single crystal Silicon
By Guixia Yang et al.

In this manuscript the authors studied the enhanced low-neutron-flux sensitivity (ELNFS) effect in silicon and its relationship with the doping concentration. The paper is rather interesting and well organized. It could be published on this journal but after several revisions according to the list of the following critical points:

1. The abstract should be more concise and synthetize the results presented in the article

2. Line 177: “3.1. Relationship between the SV of silicon with different doping concentrations and neutron flux”. We guess that the use of the acronym SV in the title of section 3.1. is inappropriate. Although later in the text the quantity Sv is introduced for the voltage noise power spectral density for silicon, we guess that the authors should reformulate the title of the section 3.1. avoiding the use symbols or acronyms.

3. Fis.2. The authors should clarify the legend of the 40 different curves in the four figures.

4. Figure 7a. The red cont. line passing through the data doesn’t correspond to any theoretical expectation or fit. We suggest to remove the cont. line and provide a theoretical expectation

Round 2

Reviewer 1 Report

The paper in its current form has been substiantially revised by the authors.
My concerns have been addressed point by point.
I want to thank the authors for the detailed answering of my questions.

For me, the only remaining unsatisfactory point is concerning Figs. 2 and 3.
The 40 different curves should be explained by a suitable legend, or at least by some additional lines in the figure caption.
I think this is not clear to the reader without additional information. From the authors detailed answer, I now understand their intention. However, explaining the nature of the 40 different curves in the subscription is I strongly suggested to improve the reader's comprehension.
